# Backbone Effects on the Thermoelectric Properties of Ultra-Small Bandgap Conjugated Polymers

**DOI:** 10.3390/polym13152486

**Published:** 2021-07-28

**Authors:** Dexun Xie, Jing Xiao, Quanwei Li, Tongchao Liu, Jinjia Xu, Guang Shao

**Affiliations:** 1School of Chemistry, Sun Yat-sen University, Guangzhou 510275, China; xiedx9@mail.sysu.edu.cn (D.-X.X.); xiaoj69@mail.sysu.edu.cn (J.X.); liquanwei@aneochem-tech.com (Q.L.); 2 Shenzhen Research Institute, Sun Yat-sen University, Shenzhen 518057, China; 3Shenzhen Key Laboratory of Polymer Science and Technology, College of Materials Science and Engineering, Shenzhen University, Shenzhen 518060, China; liu21645@163.com; 4Weldon School of Biomedical Engineering, Purdue University, West Lafayette, IN 47906, USA; xu.jinjia515@hotmail.com

**Keywords:** organic thermoelectric materials, conjugated polymers, ultra-small bandgap, carrier mobility

## Abstract

Conjugated polymers with narrower bandgaps usually induce higher carrier mobility, which is vital for the improved thermoelectric performance of polymeric materials. Herein, two indacenodithiophene (IDT) based donor–acceptor (D-A) conjugated polymers (PIDT-BBT and PIDTT-BBT) were designed and synthesized, both of which exhibited low-bandgaps. PIDTT-BBT showed a more planar backbone and carrier mobility that was two orders of magnitude higher (2.74 × 10^−2^ cm^2^V^−1^s^−1^) than that of PIDT-BBT (4.52 × 10^−4^ cm^2^V^−1^s^−1^). Both exhibited excellent thermoelectric performance after doping with 2,3,5,6-tetrafluoro-7,7,8,8-tetracyanoquinodimethane, where PIDTT-BBT exhibited a larger conductivity (0.181 S cm^−1^) and a higher power factor (1.861 μW m^−1^ K^−2^) due to its higher carrier mobility. The maximum power factor of PIDTT-BBT reached 4.04 μW m^−1^ K^−2^ at 382 K. It is believed that conjugated polymers with a low bandgap are promising in the field of organic thermoelectric materials.

## 1. Introduction

The emergence of thermoelectric materials has enabled the possibility of transforming waste heat into a usable form of energy, whereby thermal energy is directly converted into electrical energy [1,2,3]. Currently, inorganic thermoelectric materials have been extensively developed [4]; however, the toxicity and high cost of the raw materials used to develop inorganic thermoelectric materials greatly hinder their practical application [5]. In contrast, organic thermoelectric materials (OTEs) have a variety of advantages, such as facile processing, cheap raw materials, environmental friendliness, and mechanical flexibility. OTEs have developed rapidly as new types of thermoelectric materials [6,7,8].

The performances of thermoelectric materials are typically evaluated by a dimensionless figure of merit (*ZT = S*^2^*σ/**κ*), where *S*, *σ*, and *κ* are the Seebeck coefficient (μV K^−1^), electrical conductivity (S cm^−1^), and thermal conductivity (μW m^−^^1^K^−^^1^), respectively [9,10]. To obtain a large *ZT* value, it is necessary to find ways to increase both *σ* and *S*, while simultaneously reducing *κ*. However, the three parameters typically show opposing trends [1,11]. For example, an increase in *σ* often leads to a decrease in *S* and an increase in *κ*. Concurrent fine-tuning of the three parameters remains a challenge. In addition, the *κ* of OTE is generally very low [12,13], and the power factor (*PF = S^2^σ*) is often used instead of the *ZT* value to evaluate the performance of OTEs [14].

Conjugated polymers (CPs) have been widely used in optoelectronic devices, such as organic solar cells (OSCs) [15], organic light-emitting diodes (OLEDs), and organic field-effect transistors (OFETs) [16,17,18,19,20,21]. However, the application of CPs in OTEs is still limited because untreated pristine CPs usually exhibit lower *σ* [22,23]. To optimize the performance of thermoelectric devices, chemical doping (for example, 2,3,5,6-tetrafluoro-7,7,8,8-tetracyanoquinodimethane (F4TCNQ), I_2,_ and FeCl_3_) is typically performed to adjust the thermoelectric properties of CPs [24,25,26,27]. The appropriate dopant is vital to enhance the performance of OTEs [28,29,30]. Therefore, exploring the dopant species, the doping mechanisms, and the influence of different molecular structures on thermoelectric properties help to better extend the application of CPs in the thermoelectric direction.

In this work, two kinds of donor–acceptor (D-A) conjugated polymers (PIDT-BBT and PIDTT-BBT, Figure 1) were designed and synthesized with ultra-small band gaps. With the introduction of an additional thiophene unit, PIDTT-BBT exhibited a relatively flatter conjugated skeleton, leading to a larger carrier mobility (2.74 × 10^−2^ cm^2^ V^−1^s^−1^). Based on the previous studies, a high carrier mobility improves the thermoelectric performance. Subsequently, the effects of polymer structures on thermoelectric properties and the mechanism of molecular doping were extensively investigated.

## 2. Experimental

### 2.1. Materials

M1 and M2 (Figure 1) were purchased from Suna Tech Inc. (Suzhou, China), tris(2-methylphenyl)phosphine (P(*o*-tol)_3_) and tris(dibenzylideneacetone) dipalladium(0) (Pd_2_(dba)_3_) were purchased from Greenchem Technlogy Co., Ltd. (Beijing, China). M3 was received from Derthon Optoelectronic Materials Science Technology Co., Ltd., (Shenzhen, China). Anhydrous chlorobenzene (PhCl), 1,2-dichlorobenzene, anhydrous acetonitrile and F4TCNQ were obtained from Sun Chemical Technology Co., Ltd. (Shanghai, China). Other chemical reagents (including deionized water, methanol, and acetone) were obtained from commercial sources. All reagents were in analytic grade and were used without further treatment unless otherwise noted.

### 2.2. Synthesis of Monomers and Polymers

The synthetic procedure of PIDT-BBT: The mixture of M1 (0.300 g, 0.243 mmol), M3 (0.857 g, 0.243 mmol), Pd_2_(dba)_3_ (0.011 g, 0.012 mmol) and P(*o*-tol)_3_ (0.019 g, 0.061 mmol) was added into a Schlenk flask, then purged with nitrogen and sealed. After adding PhCl (5 mL) to the flask through a syringe, the solution mixture was stirred at 110 °C for 72 h. The polymer was precipitated by the addition of excess amounts of methanol. The precipitate was sequentially washed with methanol, acetone, hexane, and deionized water. After drying under vacuum, PIDT-BBT was obtained as a black solid (0.251 g, 94.2%). ^1^H NMR of PIDT-BBT (400 MHz, C_6_D_6_, TMS, r. t.); *δ*/ppm 1.15 (44H, –CH_2_–CH_3_), 2.45 (8H, –CH_2_), 6.88–7.24 (8H, aromatic), 7.31–7.78 (8H, aromatic), 8.15 (8H, benzene), 9.41 (2H, thiophene).

PIDTT-BBT was synthesized by using similar synthetic routes and was obtained as a black solid (0.220 g, 97.9%). ^1^H NMR of PIDTT-BBT (400 MHz, C_6_D_6_, TMS, r. t.); *δ*/ppm 0.16–0.34 (12H, –CH_2_–CH_3_), 0.77–1.39 (32H, –CH_2_), 2.41 (8H, –CH_2_), 6.45–6.86 (8H, aromatic), 6.94–7.27 (8H, aromatic), 7.31–7.52 (4H, aromatic).

### 2.3. Preparation of Polymer Films

PIDT-BBT and PIDTT-BBT were dissolved in 1,2-dichlorobenzene with concentrations of 8 and 6 mg mL^−1^, respectively. The dopant (F4TCNQ) was dissolved in anhydrous acetonitrile at a concentration of 16.93 mg mL^−1^, and added into the polymer solution (the molar ratio of dopant to polymer repeating unit was 1:4). The doped polymer solution was obtained after ultrasonic dispersion for 3 h. Polymer films were obtained by solvent-casting the doped polymer solution onto glass substrates (15 mm × 15 mm, washed 30 min sequentially with deionized water, acetone, and isopropanol) under ambient conditions. The doped films were used for further testing after the solvent evaporated in ambient conditions.

### 2.4. Instruments and Measurements

^1^H NMR spectra of the polymers were acquired on Agilent Technologies 400 MHz NMR spectrometer in C_6_D_6_. The molecular weights of the polymers and the polymer dispersity index (PDI) were determined via gel permeation chromatography (GPC) (Waters, Milford, MA, USA) using THF as an eluent. Thermal gravimetric analysis (TGA) was performed on TGA-55 (TA Instruments, New Castle, DE, USA) from room temperature to 600 °C under a nitrogen flow with a heating rate of 10 °C min^−1^. Differential scanning calorimetry (DSC) was performed on DSC7020 (Hitachi, Tokyo, Japan). Ultraviolet–visible (UV–vis) absorption spectra were acquired using a PerkinElmer Lambda 950 spectrophotometer. The morphology and thickness of the polymer films were observed using a scanning electron microscope (SEM) (Hitachi SU-70 system) and a contact surface topography-measuring instrument (SURFCORDER ET 4000M). Tapping-mode atomic force microscopy (AFM) images of the polymer films were obtained by using AFM (Bruker Dimension ICON) to observe the roughness of the films’ surfaces. The cyclic voltammetry (CV) was performed on a CHI 660E electrochemical workstation, and a platinum plate was used as a working electrode, Ag/Ag^+^ was used as a reference electrode, and a platinum wire was used as counter electrode in 0.1 M tetrabutylammonium hexafluorophosphate (Bu_4_NPF_6_) acetonitrile solution under a nitrogen atmosphere and scan rate of 50 mV s^−1^. The reference electrode was calibrated with the ferrocene/ferrocenium (Fc/Fc^+^) couple. Grazing incidence X-ray diffraction (GI-XRD) was measured on an X-ray diffractometer (SmartLab, Tokyo, Japan) with a copper target (λ = 1.54 Å), and the incident range was 2°–50°. X-ray photoelectron spectrometer (XPS) data were obtained using a field emission auger spectrometer (Thermo Fisher ESCALAB 250X). The *σ* and *S* of the polymer films were collected using an MRS-3 thin-film thermoelectric test system (Wuhan Joule Yacht Science & Technology Co., Ltd., Wuhan, China).

## 3. Results and Discussion 

PIDT-BBT and PIDTT-BBT were successfully synthesized via Stille coupling of two monomer units. The two polymer structures were characterized via ^1^H NMR spectroscopy (Appendix A), where all peaks were correlated and assigned to the structures of the polymers. The number-average molecular weight (*M_n_*) values of PIDT-BBT and PIDTT-BBT were 16.85 kDa and 16.76 kDa, respectively, with PDI of 2.63 and 2.38, respectively (Appendix A). The thermal stabilities of the two polymers were obtained via TGA, as shown in Appendix A. The 5% weight loss temperatures (*T*_d_) of PIDT-BBT and PIDTT-BBT were estimated to be 411 °C and 416 °C, respectively. Both polymers showed good thermal stability, and the additional thiophene ring on the PIDTT-BBT backbone does not have any significant effects on the thermal stability of the polymer. From DSC results (Appendix A), neither endothermic nor exothermic processes were observed in the temperature range from 0 °C to 300 °C, indicating that both polymers were amorphous. The molecular weights and thermal stabilities of the two polymers are summarized in Appendix A.

The energy levels of the two conjugated polymers obtained from cyclic voltammetry (CV) and UV-vis-NIR absorption spectroscopies are described in Figure 2. The onset oxidation and reduction potentials (*E*_ox_ and *E*_red_) of PIDT-BBT and PIDTT-BBT were estimated to be 0.21/−0.59 V and 0.32/−0.64 V, respectively. The lowest unoccupied molecular orbital energy levels (*E*_LUMO_) and the highest occupied molecular orbital energy levels (*E*_HOMO_) of PIDT-BBT and PIDTT-BBT were calculated from the onsets of the oxidation and reduction potentials, respectively. The highest occupied molecular orbital (HOMO) energy level can be calculated from *E*_ox_ by *E*_HOMO_ = −(*E*_ox_ + 4.8 − *E*_1/2_) eV [31], while the lowest unoccupied molecular orbital (LUMO) energy level can be calculated from *E*_red_ by *E*_LUMO_ = −(*E*_red_ + 4.8 − *E*_1/2_) eV. Therefore, the *E*_HOMO_ and *E*_LUMO_ levels of PIDT-BBT and PIDTT-BBT were calculated to be −4.90/−4.11 eV and −4.92/−4.06 eV, respectively. The electrochemical band gaps (*E*_g_^ec^) of PIDT-BBT and PIDTT-BBT were calculated to be 0.80 eV and 0.86 eV, respectively, indicating that both polymers have ultra-small bandgaps (Table 1).

The UV-vis absorption spectra of each polymer from casted film are presented in Figure 2d. Both polymers showed similar absorption bands at 660–1500 nm in both solution and casted film. The absorption maximum PIDTT-BBT (1123 nm) was 50 nm red-shifted compared to that of PIDT-BBT (1073 nm), which was attributed to the planarity of backbone generated from the additional thiophene ring. According to the onset of absorption maxima of the films for PIDT-BBT and PIDTT-BBT, the optical band gaps (*E*_g_^opt^) were calculated to be 0.87 eV and 0.95 eV, respectively. The introduction of a thiophene ring onto the IDT backbone does not have significant effects on the band gap of the polymers. Both polymers exhibited narrower bandgaps, primarily due to the planarity and stiffness of the IDT backbone that enabled an effective conjugation length of the polymer. To support this argument, quantum chemistry calculations were performed on PIDT-BBT and PIDTT-BBT dimers using density functional theory (DFT) at the B3LYP/6-31G* (d, p) level with Spartan software model 2016, in which the alkyl chain was replaced with a methyl group for clarity (Appendix A). The HOMO surface of the PIDT-BBT dimer was generally delocalized on the benzene ring of the side chain, while the HOMO surface of the PIDTT-BBT dimer was distributed on both the benzene ring of the side chain and the main chain, mainly due to the introduction of the strong electron donating group (the thiophene ring) into the PIDTT-BBT backbone. The bandgaps (*E*_g_) of the PIDT-BBT and PIDTT-BBT dimers from DFT calculations are the same (0.6 eV). These results indicate that the introduction of a thiophene ring onto the backbone of PIDTT-BBT does not induce changes in the band gap of the polymer. To investigate the mobility difference, organic field effect transistor (OFET) devices (detailed information in the Appendix A) were fabricated. The hole mobility of PIDTT-BBT was two orders of magnitude higher than that of PIDT-BBT (Table 1), which has been shown to enhance the *σ* of semiconducting polymers.

As shown in Appendix A, the *σ* (Appendix A) and *S* (Appendix A) of the polymer films doped with varying F4TCNQ concentrations were measured. As the F4TCNQ concentration increased, both polymers showed an increasing trend in *σ* and a decreasing trend in *S*, which were primarily caused by the increased carrier concentration. The polymer film with 25 mol% doping concentration exhibited the largest *PF* and was used for the subsequent thermoelectric performance testing (Appendix A). It should be noted that PIDTT-BBT films exhibited much larger values of thermoelectric properties as compared to that of the PIDT-BBT films at a specified concentration of F4TCNQ. Figure 3 describes the temperature-dependent *σ*, *S*, and *PF* of PIDT-BBT and PIDTT-BBT films doped with F4TCNQ (25 mol%) in the temperature range of 398–418 K. PIDTT-BBT films exhibited larger thermoelectric performance than that of PIDT-BBT under F4TCNQ doping conditions. As shown in Figure 3a, the *σ* values of PIDT-BBT and PIDTT-BBT films are 0.096 and 0.181 S cm^−1^ at room temperature, respectively. With increasing the temperature, the *σ* reached maximum values of 0.203 S cm^−1^ (PIDT-BBT) and 0.367 S cm^−1^ (PIDTT-BBT) at 382 K, respectively. Further increases in temperature led to a decrease in *σ*, while the *S* values showed an opposite trend, as the increase in *σ* is largely due to the increase in carrier concentration. As shown in Figure 3c, PIDTT-BBT films achieved a maximum *PF* of 4.04 μW m^−1^ K^−2^ at 382 K, and a maximum *PF* of 1.47 μW m^−1^ K^−2^ for PIDT-BBT films was obtained at 400 K. The main reason is that the pristine PIDTT-BBT films exhibited hole mobility two orders of magnitude higher than that of the pristine PIDT-BBT films.

Figure 4a,b show the GI-XRD curves of pristine and doped films. The pristine films of both polymers, PIDT-BBT and PIDTT-BBT, showed broad diffraction peaks at 22.3° and 23.7°, respectively, which can be attributed to the π-π* stacking of the polymer backbones. The π-π* stacking distances of PIDT-BBT and PIDTT-BBT are calculated to be 3.97 Å and 3.75 Å, respectively. The small stacking distance of PIDTT-BBT should be attributed to the introduction of the thiophene ring into the backbone of PIDTT-BBT, enabling a more planar backbone. No characteristic additional peaks appeared after polymer doping, indicating that the polymers did not undergo structural change after F4TCNQ doping. Elemental identification in both PIDT-BBT and PIDTT-BBT (C, N, and S) by XPS spectra is shown in Figure 4c,d. A new characteristic peak of F1s appeared after doping, which is derived from the F4TCNQ dopant. Next, the dispersion of the dopant in the polymer film was observed through the EDS diagram (Appendix A), and it was found that the F4TCNQ was evenly distributed in the films without any agglomeration.

UV-vis-NIR spectra can be used as a probe to explore the doping mechanism of polymers. As shown in Figure 5a,b, PIDT-BBT and PIDTT-BBT films exhibited strong absorption peaks at 650–1330 nm after doping, primarily due to the interactions generated through π-π* stacking of the polymer backbones [32]. However, after doping with F4TCNQ, both PIDT-BBT and PIDTT-BBT exhibited new absorption peaks in the long wavelength range, which indicated that charge transfer occurred to generate polarons [33]. This result demonstrated that F4TCNQ is an effective dopant for the two polymers. Ultraviolet photoemission spectroscopy (UPS) measurements were used to investigate valence electrons and extract work functions of the polymers. The Fermi levels, with respect to vacuum, were obtained from the difference of the intercept of the trailing edge of the secondary electron onset for PIDT-BBT and PIDTT-BBT films after doping with F4TCNQ, as shown in Figure 5c (the He(I) excitation energy is 21.2 eV). The work functions of PIDT-BBT and PIDTT-BBT pristine films were calculated to be 3.67 and 4.21 eV, respectively, and the work functions of the two polymers increased significantly to 4.63 and 4.93 eV, respectively, after doping. It was found that the Fermi level of the polymer was shifted to a lower binding energy after doping, which is equivalent to the movement of the Fermi level in the HOMO direction (Figure 5d). These results demonstrated the generation of hole carriers and effective p-type doping [34]. In addition, the lower Fermi level indicates high band degeneration because carriers can be distributed into more bands [35], which results in a higher *σ*.

As shown in Figure 6, atomic force micrographs (AFM) and scanning electron microscopy (SEM) (Appendix A) were used to investigate changes in the film morphology and the aggregation state of the dopant in the film. From the SEM images, the surfaces of the two pristine polymer films were relatively smooth, indicating that the larger molecular weight of the polymer also enabled good film processing properties. In addition, due to the strong π-π* stacking of the polymer backbone [36], the polymer surface exhibited a regular arrangement of layered structures. This phenomenon was also confirmed via cross-sectional SEM images of the pristine polymer films (Appendix A). The root mean square (RMS) of the PIDTT-BBT (6.36 nm) pristine films is larger than that of PIDT-BBT films (1.22 nm) because of the more rigid main chain structure of PIDTT-BBT. From the SEM images, it was found that the dopant did not significantly affect the film morphology. From the AFM image, no dopant aggregation was found. The RMS of PIDTT-BBT film became lower after doping, which proved that F4TCNQ showed better doping effects on PIDTT-BBT than on PIDT-BBT.

## 4. Conclusions

In this work, two D-A polymers with ultra-narrow band gaps, PIDT-BBT and PIDTT-BBT, were synthesized. With the introduction of an additional thiophene ring into the polymer backbone, the hole mobility increased by two orders of magnitude (from 4.52 × 10^−4^ to 2.74 × 10^−2^ cm^−2^ V^−1^ s^−1^), which is important for the *σ* improvement of semiconducting polymers. Improved thermoelectric performance was achieved via molecular doping with F4TCNQ via solution mixing, and PIDTT-BBT films exhibited a higher *σ* of 0.181 S cm^−1^ compared to PIDT-BBT (0.096 S cm^−1^) at room temperature. A maximum *PF* of 4.04 μW m^−1^ K^−2^ at 382 K was obtained for PIDTT-BBT films, which was significantly higher than that of PIDT-BBT films (1.47 μW m^−1^ K^−2^ at 400 K). The introduction of strong electron-donating thiophene units into the polymer backbone showed increased carrier mobility. It is believed that this work can be a useful guide to design thermoelectric materials with improved performance.

## Figures and Tables

**Figure 1 polymers-13-02486-f001:**
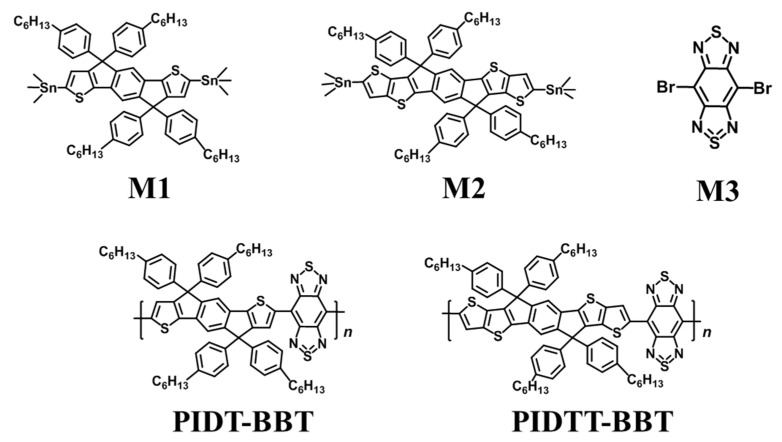
Chemical structure of the monomer and polymers.

**Figure 2 polymers-13-02486-f002:**
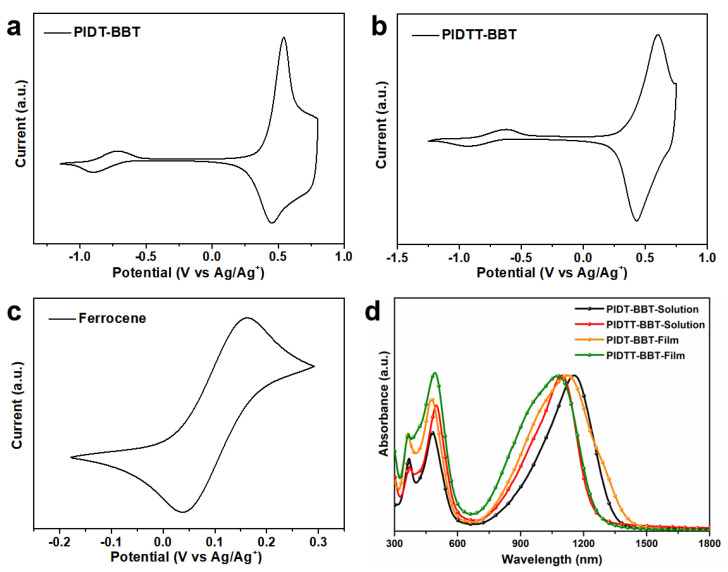
Cyclic voltammograms of solvent-casted (**a**) PIDT-BBT, (**b**) PIDTT-BBT films; (**c**) cyclic voltammograms of ferrocene solutions, and (**d**) UV-vis-NIR absorption spectra of pristine PIDT-BBT and PIDTT-BBT in solution and cast film.

**Figure 3 polymers-13-02486-f003:**
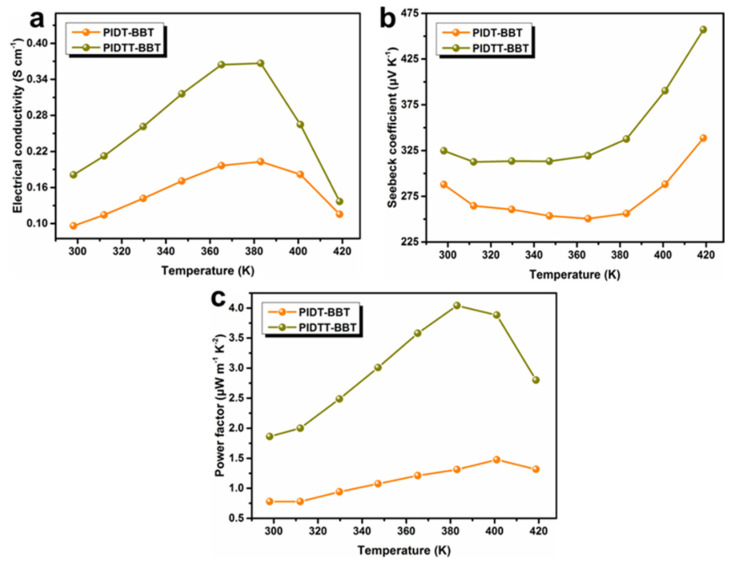
Temperature dependence of (**a**) *σ*, (**b**) *S* and (**c**) *PF* for PIDT-BBT and PIDTT-BBT films doped with F4TCNQ (25 mol%) from 298 K to 418 K.

**Figure 4 polymers-13-02486-f004:**
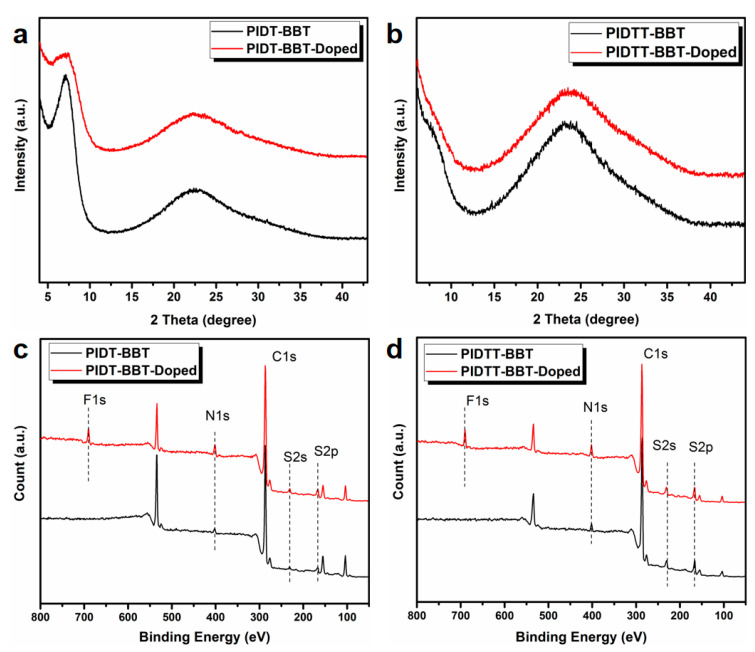
GI-XRD curves of pristine and doped (**a**) PIDT-BBT and (**b**) PIDTT-BBT films; XPS spectra of pristine and doped (**c**) PIDT-BBT and (**d**) PIDTT-BBT films.

**Figure 5 polymers-13-02486-f005:**
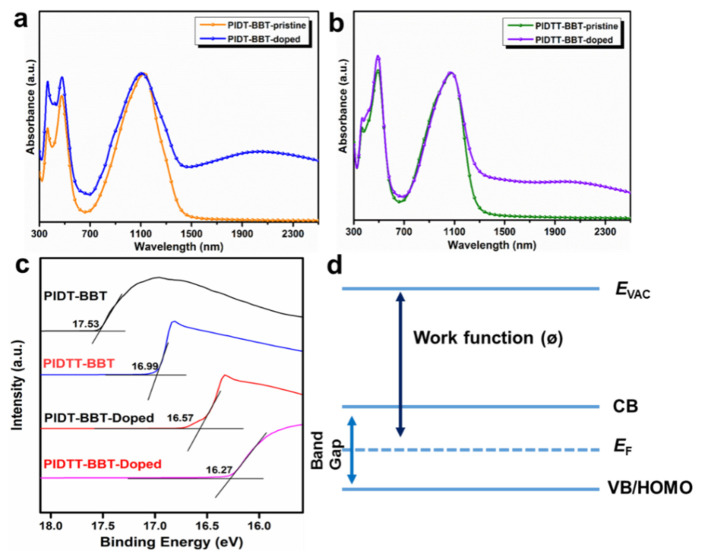
UV-vis-NIR absorption spectra of (**a**) PIDT-BBT and (**b**) PIDTT-BBT films after doping; (**c**) UPS spectra for the secondary electron cut-off region of polymers; (**d**) semiconductor level structure diagram.

**Figure 6 polymers-13-02486-f006:**
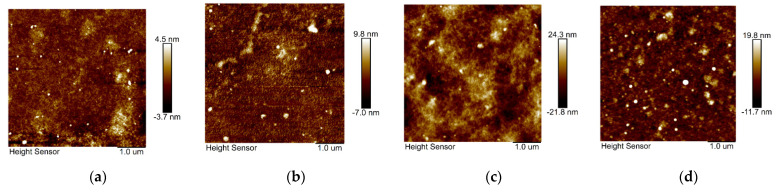
AFM height images of PIDT-BBT (**a**) pristine films and (**b**) films doped with F4TCNQ, PIDTT-BBT (**c**) pristine films and (**d**) films doped with F4TCNQ.

**Table 1 polymers-13-02486-t001:** Electrochemical properties, optical properties, and carrier mobilities of the polymers.

Polymer	*E*_HOMO_ (eV)	*E*_LUMO_ (eV)	*E*_g_^ec^ (eV)	*E*_g_^opt^ (eV)	λ_onset_ (nm)	*E*_g_^DFT^ (eV)	Carrier Mobility (10^−4^ cm^2^/V s)
PIDT-BBT	−4.90	−4.11	0.80	0.87	1423	0.60	4.52
PIDTT-BBT	−4.92	−4.06	0.86	0.95	1300	0.60	274

## Data Availability

The data presented in this study are available on request from the corresponding author.

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
