# Peer review of "Backbone Effects on the Thermoelectric Properties of Ultra-Small Bandgap Conjugated Polymers"

_polymers, 2021, doi:10.3390/polym13152486_

Round 1
Reviewer 1 Report
This manuscript describes synthesis and characterization of new polymers with low band gap. The electronic structures were analyzed with variety of techniques focusing on thermoelectric properties. The results are very useful to the readers. One problem: Details of the FET measurement is not found in the supporting info, although it is mentioned in line 176.
Reviewer 2 Report
Dear authors,
first of all, I would like to thank you for submitting your manuscript to the Polymers journal.
The article deals with synthesis and analysis of two kinds of donor-acceptor conjugated polymers, namely PIDT-BBT and PIDTT-BBT. Both of them are characterized with ultra-small band gaps.
The aim of the research is to increase carrier mobility of conjugated polymers that is beneficial for improving their thermoelectric performance. So, the effects of polymer structures on thermoelectric properties and the mechanism of molecular doping were investigated. The topic of the manuscript is very interesting and seems to be industrially important because conjugated polymers have been widely used in various opto-electronic applications, such as organic solar cells, organic light-emitting diodes and organic field-effect transistors. In addition, advanced experimental techniques, for instance, nuclear magnetic resonance (NMR), gel permeation chromatography (GPC), thermal gravimetric analysis (TGA), differential scanning calorimetry (DSC), scanning electron microscopy (SEM), atomic force microscopy (AFM), cyclic voltammetry (CV), grazing incidence X-ray diffraction (GI-XRD) and X-ray photoelectron spectroscopy (XPS) were used for the evaluation of polymers structural, morphological and thermoelectric properties.
The introduction provides sufficient theoretical background. All references used in the manuscript are relevant. Methods used during the experiment are described adequately and in detail. So, the experiment could be easily repeated. The structure of experiment is defined in logical sequence. With a few exceptions, results are clearly presented and supports achieved results. Based on above, I have only few additional questions, or suggestions for improvements. So, I recommend to publish the article after minor revision.
In line 36 authors stated that „the performances of TE materials are typically evaluated by a dimensionless quantity called Fig. of merit”. I think that, it would be clearer for readers to give full name of the quantity “figure of merit”, because “Fig.” could be falsely referred to a Figure. I also recommend to mention there units of Seebeck coefficient, electrical conductivity and thermal conductivity.
Line 54 and 55 - please, explain the difference in designation of both donor-acceptor conjugated polymers (PIDT-BBT and PIDTT-BBT). From the manuscript, it is obvious that IDT is an abbreviation of indacenodithiophene, but what does BBT mean and what is the difference between PIDT and PIDTT?
Line 138 - In Figure 2, the Y-axis is named “Courrent”, but the correct term is “Current”.
Why authors designate the number of figures by numerals, but within the text, references of figures are designated also by a letter “s”?
In line 253 authors stated that “The polymer surface exhibited regular arrangement of layered structures, which was also confirmed by cross-sectional SEM images of the pristine polymer films (Fig. S8)”. But I miss the polymer films cross-sections in the manuscript. Please, add it there.
Best regards
